# Antioxidant Nutraceutical Strategies in the Prevention of Oxidative Stress Related Eye Diseases

**DOI:** 10.3390/nu15102283

**Published:** 2023-05-12

**Authors:** Umberto Rodella, Claudia Honisch, Claudio Gatto, Paolo Ruzza, Jana D’Amato Tóthová

**Affiliations:** 1Fondazione Banca degli Occhi del Veneto Onlus (FBOV), 30174 Zelarino, Italy; urodella@alchimiasrl.com; 2Research and Development, AL.CHI.MI.A. S.R.L., Viale Austria 14, 35020 Ponte San Nicoló, Italy; cgatto@alchimiasrl.com; 3Institute of Biomolecular Chemistry of CNR (ICB-CNR), Via F. Marzolo, 1, 35131 Padova, Italy; c.honisch@icb.cnr.it

**Keywords:** antioxidants, redox balance, food supplements, nutraceuticals, ocular diseases, oxidative stress, ageing

## Abstract

This review aims to discuss the delicate balance between the physiological production of reactive oxygen species and the role of antioxidant nutraceutical molecules in managing radicals in the complex anatomical structure of the eye. Many molecules and enzymes with reducing and antioxidant potential are present in different parts of the eye. Some of these, such as glutathione, N-acetylcysteine, α-lipoic acid, coenzyme Q10, and enzymatic antioxidants, are endogenously produced by the body. Others, such as plant-derived polyphenols and carotenoids, vitamins B2, C, and E, zinc and selenium, and omega-3 polyunsaturated fatty acids, must be obtained through the diet and are considered essential nutrients. When the equilibrium between the production of reactive oxygen species and their scavenging is disrupted, radical generation overwhelms the endogenous antioxidant arsenal, leading to oxidative stress-related eye disorders and aging. Therefore, the roles of antioxidants contained in dietary supplements in preventing oxidative stress-based ocular dysfunctions are also discussed. However, the results of studies investigating the efficacy of antioxidant supplementation have been mixed or inconclusive, indicating a need for future research to highlight the potential of antioxidant molecules and to develop new preventive nutritional strategies.

## 1. Introduction

Energy production in aerobic organisms mainly depends on oxygen metabolism, occurring in mitochondria. Oxygen is abundant within cells and readily accepts free electrons generated by normal metabolism generating various kinds of reactive oxygen species (ROS), such as the superoxide anion radical (O_2_^•−^), hydrogen peroxide (H_2_O_2_), and the hydroxyl radical (HO^•^).

Additionally, ROS may be produced by other cellular components, including enzymes such as xanthine oxidase, cyclooxygenases, lipoxygenases, and the cytochrome P450-system [1].

Under physiological and relatively low concentrations, ROS participates in intra- and inter-cellular signalling pathways inducing reversible post-translational modifications of critical regulatory elements, which, in turn, control many cellular functions, including proliferation, differentiation, migration, and survival [2,3]. On the other hand, excessive ROS generation leads to uncontrolled reactions with other molecules, such as lipids, carbohydrates, and proteins, altering their functionality.

To control ROS production and to keep their levels at relatively low and safe concentrations, cells are endowed with an endogenous antioxidant system that comprises both enzymatic and non-enzymatic components [4]. These molecules shield against ROS, preventing the latter from oxidizing other biological molecules. For example, superoxide dismutase (SOD, a metalloprotein that catalyses the dismutation of superoxide radical into hydrogen peroxide and molecular oxygen), catalase (CAT, a hemoprotein deputed to the conversion of peroxides into water and oxygen), peroxiredoxins, glutathione peroxidases (GPXs, a family of selenoproteins that exploit GSH as electron donor to allow the reduction of hydrogen peroxide to alcohol and water), glutathione reductase (GR) and the thioredoxin system are among the enzymatic components of the endogenous antioxidant system [5]. Non-enzymatic components are represented by a collection of antioxidant molecules acting at different cellular compartments, such as ascorbic acid (vitamin C), alpha-tocopherol (vitamin E), retinol (vitamin A), molecules containing thiol groups (glutathione -GSH- and lipoic acid), as well as lipophilic substances (such as Coenzyme Q10 and polyunsaturated omega-3 fatty acids).

The present review focuses on the role of antioxidant molecules in nutraceutical formulations available in ocular health and oxidative stress-related disease prevention.

## 2. The Structure of the Eye and Ocular Antioxidant Defense System

The eye is a very complex organ of the human body that can be divided into the anterior and the posterior segments [6]. In the following chapters, the different molecules composing the antioxidant arsenals of the different anatomical compartments of the eye are presented (Figure 1).

### 2.1. Antioxidants Arsenal in the Anterior Segment

The anterior segment includes the ocular surface, the anterior chamber, the iris, and the lens. The ocular surface, which represents the interface towards the environment, provides cornea, conjunctiva, lacrimal and meibomian glands, nasolacrimal duct and eyelids [26]. The ocular surface is humectated by the tear film, a watery environment rich in electrolytes and proteins, resulting in a muco-aqueous lubricant gel; the outermost layer of the tear film is composed of lipids (meibum), which prevents evaporation.

Different antioxidant molecules were reported in human tears, with ascorbic acid being the most abundant (665 μM) and uric acid the second (328 μM). These two compounds account for ~50% of the total antioxidant activity; other small molecules, including GSH (107 μM), L-cysteine (48 μM) and L-tyrosine (45 μM) make up the rest [7]. Zinc and selenium were also identified in the tear film [8]. However, the only reported antioxidant enzyme in the tear film is SOD [9]. On the contrary, many enzymatic antioxidants have been documented in the cornea. These include SODs, CAT, GPXs, GR, glucose-6-phosphate dehydrogenase (G6PD), along with members of the aldehyde dehydrogenase (ALDH) superfamily [10].

Among the non-enzymatic antioxidants in the cornea, ascorbic acid, commonly known as vitamin C, is an excellent antioxidant agent that can donate a proton and form a relatively stable ascorbyl free radical. In addition, vitamin C was shown to be an effective scavenger of oxygen and nitrogen oxide species, such as superoxide radical ion, hydrogen peroxide, hydroxyl radical, and singlet oxygen. Its oxidation gives rise to dehydroascorbate, which is then reconverted to ascorbate, using GSH or NADPH as electron donors [9] (Figure 2).

The ascorbate accumulation in the eye is achieved primarily by active transport through the iris-ciliary body into aqueous humour and subsequent transport into the cornea and lens.

The cornea presents millimolar concentrations of GSH (4~7 mM) that, unlike ascorbate, can be synthesized via its *de novo* biosynthetic pathway mediated by two cytosolic enzymes, glutamate-cysteine ligase, and glutathione synthetase. It represents the most abundant cellular non-protein thiol and can directly scavenge hydroxyl radicals and superoxide [11].

GSH converts superoxide (O_2_^•−^) to H_2_O_2_ while undergoing self-oxidation (GSSG). H_2_O_2_ can then be converted into water by CAT or GPXs. Finally, GSSG is recycled to GSH by GR in the presence of NADPH as an electron donor (Figure 3). This redox cycle also protects lens proteins from the action of ROS.

The crystalline lens is a highly elastic, circular, biconvex, transparent body composed of long and thin cells named lens fibers, highly rich in water-soluble proteins called crystallins [12]. These cells lack cellular organelles, neural innervation and blood vessels, increasing tissue transparency and reducing the production of ROS, also in relation to the low oxygen tension of the lens [27].

Crystallin proteins combine both structural roles and chaperone activity. Indeed, the α-isoform could bind damaged proteins, sequestering misfolded or damaged aggregation-prone proteins and preventing the formation of light-scattering species [28].

Mammalian crystallins are intrinsically resistant to photo-damage due to the presence of tryptophans (Trp), tyrosines (Tyr) and phenylalanines (Phe) in their sequences, residues capable of UV absorption and radical photochemistry [29,30]. Trp has the highest specific absorption of protein amino acids at 280 nm. It dissipates UV-B radiation through energy transfer and internal conversion, thereby protecting the structures behind the lens, particularly the retina, from UV damage [31]. Besides Trp, Cys, Tyr and His residues can participate in excited state radical chemistry that could lead to photochemical covalent crosslinking of the involved proteins [32].

Oxidative processes in the healthy eye lens are controlled by both α-crystallin proteins and physiologically occurring water-soluble antioxidants.

GSH, ascorbic acid, GR and GPx are potent endogenous radical scavengers in lens fiber cells. GSH is the most abundant, reaching about 10 mM concentration in the lens [13].

The anterior chamber, located between the iris and the cornea’s innermost surface, serves as the rear support for the cornea and is filled with a clear fluid. The aqueous humor that provides nutrition removes excretory products from metabolism, transports neurotransmitters, stabilizes the ocular structure and contributes to regulating the homeostasis of the lens and cornea [33]. In addition, ROS can be continuously generated in the aqueous humor through hydrogen peroxide, superoxide anion, singlet oxygen, and peroxyl radicals. In human aqueous humor the antioxidant defence system is represented by small molecules, including ascorbic acid (530 μM), L-tyrosine (78 μM), uric acid (43 μM), L-cysteine (14.3 μM) and GSH (5.5 μM) [14], and only trace amounts of SOD [15].

### 2.2. Antioxidants Arsenal in the Posterior Segment

The posterior segment of the eye is represented by all-optical structures behind the lens, including the vitreous, retina, choroid and optic nerve.

The retina is a complex structure, rich in blood vessels and a dense network of neurons that capture and process light by virtue of the visual pigments. It is a highly energy-consuming tissue that relies on abundant mitochondrial oxidative phosphorylation and aerobic glycolysis to sustain the energy demand. Indeed, the retina has the highest oxygen consumption rate (per gram of tissue) in the body [34], which can produce a considerable amount of ROS in addition to that generated by exposure to visible and UV light. The retina developed significant defence strategies against ROS to overcome this situation, including SOD and CAT enzymes, GSH, vitamin C, carotenoids (lutein and zeaxanthin) and other pigments [16]. A vital antioxidant role is also played by ocular melanin, a highly heterogeneous and complex polymer normally present in the iris, choroid and retinal pigment epithelium [17]. Melanin exerts a protective effect on ocular cells and tissues using physical and biochemical mechanisms. In the anterior segment of the eye globe, it mainly acts as a photo-screen, while in the posterior segment, the mechanism appears to be predominantly biochemical [35]. It was suggested to be a weak antioxidant and free radical scavenger. Still, most likely, it is involved in the inactivation of the autofluorescent membrane-bound lipofuscin, an essential source of oxidative stress since it absorbs light and generates light-induced ROS [36].

The vitreous is a viscoelastic gel in the eye’s posterior chamber, with a high water content (99%) and few resident cells [37]. It occupies about 80% of the eye globe volume. It presents a vast array of both an enzymatic and non-enzymatic antioxidant profile, including SOD, GPX and CAT, as well as GSH, metal-chelating proteins (such as transferrin), uric acid, ascorbic acid, riboflavin and trace metals (zinc and selenium) [38].

Among metals crucial for the antioxidant system, zinc protects sulfhydryl groups from oxidation and competes with iron and copper ions for binding to cell membranes and proteins, displacing these potential ROS-generating metals; in addition, zinc increases the activation of antioxidant proteins, such as GSH and SOD [39]. Another essential metal is selenium, which acts indirectly through incorporation in selenoproteins having antioxidant properties.

## 3. Oxidative Stress and Eye Pathologies

When antioxidant patrolling is inadequate and/or ROS production is excessive, the delicate balance between ROS generation and ROS scavenging executed by antioxidants leads to “oxidative stress”, a chronic condition in which ROS overwhelm antioxidant shield activity. In this scenario, progressive cell dysfunctionality follows the uncontrolled reaction of ROS with cellular proteins, lipids, and other cellular molecules.

Oxidative stress has been implicated in aging and a variety of age-related chronic diseases, including several eye pathologies, such as Dry Eye Disease (DED), cataracts, glaucoma, Age-related Macular Degeneration (AMD) and Diabetic Retinopathy (DR) [40].

DED is a disorder involving the ocular surface, in which an excess of inflammatory cytokines increases ROS production, disrupting the healthy tear film [41].

A cataract occurs when the ordered arrangement of crystallin proteins composing the scaffold of the lens is lost due to excessive exposure to UV radiation and ROS, and crystallins aggregation, consequent to their misfolding, leads to lens opacification and progressive vision worsening.

AMD is a common cause of progressive visual impairment in the elderly caused by oxidative degeneration of the retina [42].

Glaucoma is a group of eye disorders that damage the optic nerve. While the aetiology is multifactorial, oxidative stress is essential in disease progression.

DR is a most common complications of diabetes mellitus, whose hallmark, i.e., hyperglycaemia, is associated with increased accumulation of ROS in different tissues, including the retina [43].

## 4. Nutraceutical Antioxidants for the Ophthalmic Field

The correlation between oxidative stress, oxidative damage and ocular disease has led researchers to investigate the antioxidant approach in modulating ophthalmic pathologic conditions. The reinforcement of the endogenous antioxidant system of ocular tissues can be achieved through nutrition and food supplementation, as well as through medical devices and antioxidant-based drugs. Although they may contain the same antioxidant molecule(s), nutraceuticals, medical devices, and drugs require different market approval routes and have other purposes. Therefore, we investigated the presence of antioxidants in the ingredient lists of nutraceuticals supplements intended to be used in ophthalmology. If available, we also opted to include studies from non-enteral routes of administration (such as antioxidant(s)-based eye drops), as the goals of these treatments are comparable to the oral supplementation, i.e., delivering one or more antioxidant molecules to the site of action (eye) and eliciting one or more antioxidant-related effect.

On the other hand, since our principal aim was to identify and investigate the antioxidant properties of substances with a defined chemical formula (single molecules), we decided not to include botanical extracts in this review. Indeed, although many botanical extracts have antioxidant effects that could be useful in the ophthalmic field, they have a chemical composition that is difficult to standardise. Furthermore, the molecules contained in a botanical extract are subject to multifactorial variations related to, but not limited to, the latitude and cultivation method of the plant, the seasonality of harvesting, and the extraction method.

The antioxidant molecules presented in the following sections (Table 1) will be divided into three main categories:(i)Exogenous plant-derived antioxidants: they are defined as “essential” nutrients, as they are not synthesized by the human body and, thus, can be obtained exclusively through exogenous introduction into the body (usually nutrition). We sub-divided them by their chemical class:
Polyphenols.Carotenoids.(ii)Water-soluble promoters of the endogenous antioxidant system: molecules either produced or not by the human organism, showing direct antioxidant activity and/or able to boost the endogenous antioxidant system.(iii)Lipophilic antioxidants: lipophilic molecules promoting the defence of cellular membranes.

## 5. Exogenous Plant-Derived Antioxidants

### 5.1. Flavonoids

Polyphenols comprise a family of around 5000 organic molecules in the plant kingdom. In plants, they are secondary metabolites that play a role in the defence of the vegetal organism against UV radiation or pathogen aggression [49].

Among them, flavonoids, reported in various fruit and vegetables, including tea, cocoa, and red wine, are characterised by a common backbone represented by two aromatic rings bound together by three carbon atoms forming an oxygenated heterocycle (Figure 4, left).

Different mechanisms have been proposed to explain their antioxidant capacity, involving the suppression of ROS formation by either inhibiting the enzyme system deputed to the formation of oxygen radicals, such as hydrogen peroxidase and nitric oxide synthetase, directly scavenging radicals, chelating or stabilizing metal ions involved in the redox reactions, or upregulating the host antioxidant defences. *In vitro* and *in vivo* studies demonstrated that flavonoids have strong anti-inflammatory, immunomodulatory and antioxidative properties [50]. Quercetin (Figure 4, right) is one of the most abundant and well-studied flavonoids. In an experimental mouse model of dry eye disease (DED), topical application of quercetin increased tear volume, corneal regularity, and goblet cell density. This was associated with reduced inflammatory markers, including MMP-2, MMP-9, ICAM-1 and VCAM-1 in the lacrimal gland [51]. In another DED murine model, topical quercetin protected the ocular surface, enhancing corneal integrity, as highlighted by fluorescein staining [52].

Moreover, quercetin applied topically or added to the diet of galactose-fed rats (an *in vivo* model of diet-induced cataract) showed an anticataractogenic effect as it delayed lens damage. However, no direct antioxidant activity was observed in this study [53]. On the other hand, in an *ex vivo* rat model of hydrogen peroxide-induced lens opacification, quercetin, in low micromolar (10 μM) concentration, has been an effective inhibitor of oxidative damage and cataractogenesis [54]. Unfortunately, quercetin displays poor chemical stability, and it is rapidly metabolised into 3′-O-methyl quercetin by catechol-O-methyl transferase in rat lenses, reducing its antioxidant activity [55].

Flavonoids from several sources have efficiently reduced the extent of selenite cataracts in rat models [56,57,58,59].

In an advanced 3D *in vitro* study of 3D-human trabecular meshwork cells, a patented eye drop formulation containing >2.5% polyphenols from *Perilla frutescens* had significant inhibitory effects on the apoptotic pathway, activation of NF-κB, and induction of pro-inflammatory cytokines from oxidative stress. These results suggest that polyphenols-based eye drops could be an adjuvant therapy in glaucoma [60].

### 5.2. Catechins

Catechins are a class of polyphenols belonging to the flavon-3-ols subgroup and present two chiral centres (highlighted by stars in Figure 5). Naturally occurring compounds exhibit a 2,3-*trans* (catechins) or a 2,3-*cis* (epicatechin) configuration.

Reported data indicate they exert antioxidant activity, directly scavenging ROS and chelating metal ions. The phenolic hydroxyl groups can react with oxygen and nitrogen radicals, donating one electron and reducing the free radicals. In addition, catechins can take charge of the radical, which is stabilised by charge delocalisation on the π-electrons of the phenyl ring, thereby terminating the radical propagation [61]. Adjacent hydroxyl groups form metal chelation sites that sequester metals involved in free radical formation [62]. Moreover, catechins regulate the expression of several enzymes, inducing antioxidants, inhibiting pro-oxidant ones, and increasing detoxification enzymes [63].

Green tea catechins are a group of flavonoids in the leaves of *Camellia sinensis* composed of epicatechin, epigallocatechin, epicatechin gallate and epigallocatechin gallate (EGCG). In an *in vitro* study, EGCG showed to inhibit different kinds of cytokines induced by IL-1 or hyperosmolarity in a dose-dependent manner in cultured human corneal epithelial cells [64]. Additionally, EGCG decreased the expression of IL-1 in a murine model of DED and reduced corneal epithelial damage [65]. Moreover, EGCG showed antioxidant activity by efficiently preventing H_2_O_2_-mediated oxidation of cataractous human γ-crystallin [66].

Masmali and co-workers observed that green tea consumption could negatively affect the tear film quality, measured by the tear ferning test [67]. It was suggested that this effect could be due to the oxidation of the lipid layer and/or the alteration of electrolyte content [68]. On the contrary, a randomized controlled trial that evaluated the efficacy of green tea extract in 60 patients with DED secondary to meibomian gland dysfunction showed an improvement in symptoms, break-up time and meibum quality compared to the control group [69]. The contrasting data underline the need to further investigate the effects and mechanisms of action of catechins-containing extracts on tear film physiology.

### 5.3. Anthocyanins

Anthocyanins are an important group of visible plant pigments responsible for many plants and fruits’ red, blue, and purple pigmentation (Figure 6).

Supplementation with anthocyanin-rich extracts of bilberry (*Vaccinium myrtillus* L.) was evaluated in different studies. A randomized controlled trial enrolled patients with DED, and the supplementation of bilberry extract showed to significantly improve the tear volume, evaluated by the Schirmer tests, which measures the quantity of lacrimal secretion if compared to control subjects [70]. A clinical pilot trial investigated the effects of MaquiBright^®^, a registered trademark maqui berry (*Aristotelia chilensis*) extract with a standardized 35% content of anthocyanins, in patients with moderate DED, reporting a significant improvement of Schirmer test and quality of life after 60 days of treatments [71]. Similar results on the same extract were obtained in a randomized controlled trial, which reported a significant improvement in the Schirmer test and relief from symptoms of eye fatigue in the visual display terminal [72].

### 5.4. Curcumin

Curcumin is a polyphenol isolated from turmeric (*Curcuma longa*), widely used as a spice and flavouring agent. It is a linear diarylheptanoid characterized by keto-enol tautomerism (Figure 7), in which equilibrium is shifted in solution towards the -enol form, stabilized by the fully conjugated aromatic system. As such, it is extremely useful as an antioxidant since it can chelate different positively charged metal ions that often trigger the formation of radicals [73]. However, the exploitability of this molecule as a food supplement is limited by its poor solubility in aqueous solution, low absorption in the gut, rapid metabolism (with degradation to antioxidant by-products such as ferulic acid, vanillin and dehydrozingerone [74]) and systematic elimination [75]. To overcome such drawbacks, the association with piperine, one of the significant black pepper alkaloids, enhanced curcumin bioavailability [76]. Dehydrozingerone is modestly soluble in water and stable at neutral pH, unlike curcumin. Curcumin and its derivatives zingerone, dehydrozingerone, and the biphenyl analogue bizingerone have been studied for their radical scavenging ability, related to the presence of the phenolic rings, the β-diketone, the conjugated double bond system and the methoxy substituents at the phenyl ring [77].

Curcumin has been shown to modulate multiple cell signalling pathways leading to anti-inflammatory, antioxidant, anti-angiogenetic, wound healing, and antimicrobial effects [78]. It was investigated for its impact on the ocular surface system and administered by intranasal delivery of a nanomicelle formulation to overcome low oral bioavailability. It restored ocular surface homeostasis by reducing ROS production, decreasing the expression of inflammatory mediators, and increasing neurotrophic factors [79]. The intranasal treatment effectively delivered the active ingredient to the ophthalmic branch of the trigeminal ganglion. An additional potential application field of curcumin is dry eye disease, as it was observed to reduce pro-inflammatory cytokines such as IL-4 and IL-5 in the conjunctiva of mice [80]. In human corneal epithelial cells, curcumin counteracted the increased production of IL-1, IL-6 and TNF induced by hyperosmotic stress [81], resulting to be a promising candidate for the treatment of DED.

### 5.5. Resveratrol

Resveratrol is a natural polyphenol found in plant-based foods, mainly grapes and berries, with a high concentration in red wine. Resveratrol has a stilbene structure consisting of a phenolic ring connected to a resorcinol moiety by an ethylene bridge that identifies a *trans* and a *cis* isomer (Figure 8).

It is usually present in plants as *trans*-resveratrol. Still, when ingested, it is rapidly metabolised to the biologically more active dihydro resveratrol (Figure 8, right), showing a high antiproliferative effect.

Resveratrol can directly scavenge various free radicals [82] and has been shown to inhibit lipid peroxidation more effectively than vitamins C and E [83,84,85]. It also stimulates the expression of the enzymatic antioxidant defence cellular system [86].

The direct radical scavenging activity is related to the redox properties of the phenolic hydroxyl groups and their capability of delocalizing unpaired electrons across the π-system of the structure [87]. Resveratrol also undergoes one-electron electrochemical oxidation, showing two oxidation peaks, the first corresponding to the irreversible oxidation of the phenol group and the second corresponding to the irreversible oxidation of the resorcinol moiety [88].

Conversely, resveratrol showed pro-oxidant properties, leading to DNA oxidative damage in synergy with transition metal ions such as copper. It was suggested that this pro-oxidant action might inhibit tumour initiation, promotion and progression and that this could be a common chemo-preventive mechanism of plant polyphenols [87].

Resveratrol has various beneficial effects on health, including cardio-protective action upon vasculature, i.e., the ability to inhibit angiogenesis, prevent inflammation, and facilitate vaso-relaxation [89]. Since vascular impairment is involved in many ocular diseases, such as impaired blood flow and ischaemic events in age-related macular degeneration (AMD), diabetic retinopathy (DR) and glaucoma, the potential advantages of resveratrol attracted much interest in the field of ocular diseases prevention and treatment. The significant activities attributed to resveratrol on the eye include antioxidant, anti-apoptotic, anti-tumorigenic, anti-inflammatory, anti-angiogenic and vasorelaxant.

In an *in vitro* study on primary porcine trabecular meshwork cells subjected to chronic oxidative stress (40% O_2_), chronic resveratrol treatment reduced the production of intracellular ROS. It prevented the expression of glaucoma markers IL1a, IL-6, IL-8, and ELAM-1 (endothelial-leukocyte adhesion molecule 1). In addition, resveratrol inhibited the expression of the cellular senescence marker sa-β-galactosidase (sa-β-gal), typically induced by oxidative stress [90].

In a selenite-induced cataract *in vivo* model (rat), intraperitoneal (i.p.) resveratrol (40 mg/kg/day) proved to increase the levels of GSH and to lower the amount of malondialdehyde (MDA), a marker of lipid peroxidation, in rat lenses and erythrocytes. Additionally, it suppressed selenite-induced oxidative stress and cataract formation [91]. Similar results were observed in a naphthalene-induced albino rat cataract model, where the same dose as i.p. administration of resveratrol was confirmed to reduce MDA levels and it significantly retarded lens opacity, restored antioxidants (CAT, SOD, GPX, GSH), function and reduced lipid peroxidation in the lenses of the animals [92].

In a diabetic rat model, four consecutive weeks of i.p. administration of 10 mg/kg/day resveratrol significantly alleviated hyperglycaemia and promote weight loss. They simultaneously reduced the levels of oxidative markers, along with superoxide dismutase activity in the blood and retinas [93]. Additionally, oral gavage at 20 mg/kg of resveratrol suspended in 0.5% carboxymethylcellulose dissolved in 0.9% saline solution administered once a day for four weeks showed to prevent vascular damage in the retina [94].

### 5.6. Carotenoids

Carotenoids are lipid-soluble antioxidants comprising more than 600 compounds responsible for the yellow, orange, and red coloring of fruits, leaves and flowers. They belong to the tetraterpene family and are C40-based isoprenoids. In addition, they are active ROS scavengers and induce the synthesis of antioxidant enzymes.

The progenitor of carotenoids are carotenes, whose most common form is β-carotene, an important lipophilic antioxidant, able to insert into the cell membrane, acting directly as a shield against lipid peroxidation working in synergy with other antioxidants such as vitamin C and E [95]. Additionally, it is the precursor of vitamin A, a crucial element for human physiology. In the sight process, it combines with the photoreceptor’s opsin and rhodopsin in the form of 11-cis-retinal.

A subclass of carotenoids containing oxygen is named xanthophylls [96]. Among them, lutein and zeaxanthin (Figure 9) are epimers that differ only by the configuration of a single hydroxyl group [97]. They are mainly concentrated in the macula lutea, and their peak concentration appears at the centre of the fovea. However, they are also present in other ocular compartments, such as the lens, where they display protective effects against lipid peroxidation, protein oxidation, DNA damage and upregulate GSH levels. In addition, ocular carotenoids absorb light protecting eye tissues from the light-induced photochemical insult and quench free radicals by accepting their unpaired electrons, stabilizing them in their conjugated double bonds system [98].

Lutein and zeaxanthin in the human lens absorb blue light, characterized by short wavelength and high energy [20].

Such molecules can cross the blood-brain and blood-retinal barriers of mammals, including humans, and then deposit into the retina, which can play a crucial role in regulating oxidative homeostasis [99,100]. Numerous experiments and clinical trials have been conducted to elucidate the role of orally supplemented carotenoids in preventing or modulating ocular pathologies.

AREDS clinical trials showed that the oral administration of vitamin C (500 mg), vitamin E (400 IU), β-carotene (15 mg) and zinc (80 mg) was able to significantly reduce the risk of developing AMD and to maintain visual acuity better than placebo [44]. Surprisingly, the addition of lutein (10 mg), zeaxanthin (2 mg), docosahexaenoic (350 mg) and eicosapentaenoic (650 mg) acids to the formulation failed to demonstrate extra benefits (AREDS2 clinical trials) [47]. On the other hand, in other studies (reviewed by Johra and colleagues [19]), lutein and zeaxanthin proved to be protective against the development of AMD.

In 2016 Ma et al. reported that the lutein+zeaxanthin oral administration could increase the macular pigment optical density in patients affected by AMD in a dose-dependent manner [101]. Successively, Wilson et al. reported that the increase in macular pigment optical density can be achieved with doses > 10 mg/d of lutein and/or zeaxanthin [102].

Food supplementation of lutein and zeaxanthin, but not of β-carotene, reduced the risk for senile cataracts by exerting an antioxidant activity towards lens proteins, lipids, and DNA [48,103].

However, a recent meta-analysis did not show a significant association between the serum content of lutein, zeaxanthin, and other vitamin A-derivatives with DR [104].

Another xanthophyll that is very promising in the ophthalmic field is astaxanthin (ASTX, Figure 9). This naturally occurring red carotenoid pigment is produced by some microalgae, such as *Haematococcus pluvialis*. As the primary dietary source for some marine animals, such as salmon, shrimp, and lobster, it confers their flesh to the typical pink color. Carotenoids are known to insert into the cell membrane. Electron density studies showed that apolar carotenoids, such as β-carotene and lycopene, altered the packing of phospholipid acyl chains and disordered the membrane bilayer.

On the contrary, ASTX did not modify the structure of constituent lipids. ASTX displays an amphipathic nature, which orientation, differently to other carotenoids, may be influenced by membrane lipid composition (e.g., acyl chain length, degree of saturation, presence of cholesterol). ASTX presents a keto group at the C4 and C4′ positions of the terminal rings, which are thought to stabilize membrane interactions with respect to the polar terminal groups. This would allow the rest of the molecule to span the entire membrane width, fitting precisely in the polar-nonpolar-polar depth of the cell membrane [105].

The perpendicular orientation of xanthophylls in cell membranes has been shown to increase the rigidity of the lipid bilayer, where they can act as “molecular rivets” and as a “full-length” anti-ROS shield of the lipid bilayer [106]. In addition, ASTX antioxidant activity has been reported to be more than 100 times and 550 times greater than that of vitamin E against lipid peroxidation and singlet oxygen quenching [107].

Oral treatment of ASTX, along with zeaxanthin and lutein in AMD patients, preserved retinal function better and improved visual acuity. In addition, in animal models of diabetic retinopathy, ASTX preserved histological and functional outcomes of the diseases, increasing the levels of antioxidant enzymes, and reducing levels of oxidative mediators, such as NF-κB and aldolase reductase activity [108].

Moreover, the neuroprotective effect of ASTX may be usefully adopted in managing glaucoma. For example, in a mouse model in which elevated IOP was induced by cauterizing episcleral vessels, significant reduction of retinal apoptosis and oxidative markers expression and higher visual evoked potential were observed in animals receiving ASTX in their diet. More recently, experiments have been conducted in a normal IOP glaucoma murine model. Retinal ganglion cells (RGC) axonal degeneration is caused by the homozygous deletion in the glutamate/aspartate transporter (GLAST−/−) gene. In these studies, ASTX, both given orally and i.p., protected RGC from glutamate neurotoxicity and degeneration [109,110].

Other promising ASTX applications in eye disorders include the treatment of uveitis, a wide group of inflammatory conditions affecting the middle layer of the eye, and cataract prevention, even if clinical data in humans are missing [108].

Due to the low bioavailability and water solubility, topical application of ASTX might be an alternative route of administration. In a murine model of UV-B light-induced photokeratitis, ASTX eye drops protected and well-preserved the epithelium from cell death and thinning caused by photodamage [111]. More recently, topical application of ASTX-encapsulating liposomes in a rat dry eye disease model prevented the up-regulation of age-related markers and maintained epithelial integrity [112].

## 6. Water-Soluble Promoters of the Endogenous Antioxidant System

### 6.1. Zinc and Selenium

Zinc and selenium are essential micronutrients that must be supplied through the diet. Zinc plays vital roles in the human body, which can be described as catalytic, structural and regulatory functions [113]. Selenium is a fundamental building block of a small cluster of antioxidant proteins, such as thioredoxin reductases, GPXs and iodothyronine deiodinases. These enzymes act as oxidoreductases and depend on selenium availability for proper activity [114]. Selenoproteins P, a selenium transporter found in the tear film responsible for delivering selenium to the corneal epithelium, is reduced in dry eye disease [115]. In contrast, selenium concentration is increased in the tears of diabetic patients [8]. These findings suggest that altered selenium metabolism may play a role in developing or progressing oxidative stress-related eye disorders.

Noteworthy, zinc plays an essential role in the activity of enzymes in the antioxidant defence system, in the regulation of GPXs, and as a co-factor for SOD. Zinc also induces the synthesis of metallothioneins, deputed to reducing hydroxyl radicals and sequestering ROS. In addition, this mineral has been shown to stabilise cell membranes and inhibit the pro-oxidant enzyme NADPH-Oxidase [39,116].

Eye zinc levels were decreased in AMD, suggesting that zinc deficiency may lead to oxidative stress and retinal damage [117].

The AREDS study reported that the group receiving zinc supplementation in combination with antioxidant vitamins showed a bigger decrease in advanced AMD progression (~25%) and the highest prevention of loss of vision (19%) [45,46]. Furthermore, low serum zinc levels are correlated with diabetes, hypertension and microvascular complications, suggesting the idea that it was also correlated with the duration and severity of diabetic retinopathy [118], as confirmed by studies on a rat model of DR [119].

### 6.2. Ascorbic Acid

Most probably one of the most studied and well-known antioxidant molecules, vitamin C is a water-soluble nutrient of fundamental importance. It is a “universal” antioxidant, present in several aqueous fluids and tissue compartments of the body. It is a reversible reductant and antioxidant which efficiently neutralizes ROS and reactive nitrogen species, as well as peroxynitrite, nitroxide radicals, singlet oxygen and hypochlorite. It also works in synergy with other biological antioxidant such as GSH, vitamin E and flavonoids [120,121,122]. Among mammals, only humans and a few other species (Guinea pigs, some bats and some primates) have lost this ability and must obtain the essential vitamin C from their food [123].

Ascorbic acid is widespread in all eye compartments at high concentrations, suggesting an essential function of vitamin C in eye physiology and in preventing eye disorders [21]. The AREDS study showed that vitamin C supplementation significantly reduces the risk of developing AMD and maintains visual acuity [44].

In 2022, Xiong and colleagues published a cross-sectional study and meta-analysis on the association between vitamin intake and DR. The authors concluded that lower levels of ascorbic acid were significantly associated with the disease [104].

According to the high vitamin C lens content, a large body of literature also investigated the role of vitamin C in cataract prevention, leading to conflicting results [44,124,125,126,127]. This dissension suggests that a simple intervention of vitamin C supplementation might not be effective. Accordingly, plasma and humor aqueous are estimated to be saturated in humans with intakes of <250 mg of vitamins C/day [128].

A recent study found a high correlation between total antioxidant capacity and ascorbic acid concentration in aqueous humor with cataract severity, determined by cumulative dissipated energy during phacoemulsification [129].

Ascorbic acid is known to be relatively unstable as it undergoes oxidation. For this reason, more stable derivatives have been developed, such as ascorbic acid 2-O-alpha-glucoside (AA-2G), and investigated in several applications, including the ophthalmological field. For example, studies on developing chick embryos showed that AA-2G has preventive effects on hydrocortisone-induced cataract formation and inhibits the decrease in GSH lens content and the increase in lipid peroxide lens content, even more than ascorbic acid itself [130,131]. More recently, AA-2G has also been shown to be effective in topical application in solid-in-oil nanodispersions that can promote the wound healing of corneal epithelium [132].

### 6.3. N-Acetyl Cysteine and Other Cysteine Derivatives

N-Acetyl Cysteine (NAC) is the membrane-permeable acetylated form of the amino acid L-cysteine (Figure 10). It was initially used as a mucolytic agent, as it lowers the viscosity of mucous by reducing the disulphide bonds of mucoproteins.

The presence of the thiol group can donate reducing equivalents such as hydrogen, which can reinstitute losses sustained from exposure to ROS, and this is often referred to as a ‘repair reaction’ [133]. Moreover, NAC also acts as a direct antioxidant, scavenging superoxide and peroxides [134]. However, the slow chemical kinetics of NAC reaction with ROS does not support the direct scavenging narrative. Given that NAC indeed exerts antioxidative (reductive) effects inside cells, it is supposed to do so indirectly [135].

NAC can be hydrolyzed into cysteine, a precursor in the synthesis of GSH. Topical administration of NAC has been reported to improve corneal epithelial healing of different kinds of wounds in animal and human studies, particularly alkaly burns of the ocular surface [136,137]. Additionally, NAC improves the clinical outcome of DED symptoms by having favorable lubricant activity [136]. A 4 times daily administration of either 1% or 0.1% topical NAC solution induced a statistically significant higher cone cell density, as observed by confocal microscopy [136].

The role of NAC was also investigated in cataract development in diabetic rats. NAC topical administration (0.01% and 0.05%) twice daily for 13 weeks resulted in delayed onset of early diabetic cataracts. However, this treatment did not seem to delay or reverse the severity of cataracts at later stages of the disease, as suggested by a study on an *in vitro* rabbit lens model of hyperoxia-induced cataracts [138]. On the other hand, NAC treatment delayed the progression of cataract onset and increased GSH levels in an *ex vivo* rabbit model of hyperoxia-induced lens damage [139]. *In vivo* studies on Wistar rat’s selenite-induced cataracts model assessed the activity of a different modified NAC-related compound, particularly N-acetylcysteine amide (NACA) eye drops reduced the severity of cataract formation along with increasing GSH levels [140].

Additionally, *in vitro* experiments on human retinas obtained from donors with or without AMD showed NAC to protect retinal pigment epithelium (RPE) cells in the retina from oxidative damage [141].

NAC was also investigated in the prevention of retinitis pigmentosa, a family of inherited diseases having in common a progressive retinal degeneration. A study detected improved retinal cone photoreceptor function in patients with moderately advanced RP after six months of NAC oral administration [142].

Notably, other cysteine derivatives also possess antioxidant properties that have been tested in ophthalmological applications. This is the case with S-allyl-L-cysteine (SAC), an organosulfur constituent of garlic, and S-allylmercapto-N-acetylcysteine (ASSNAC). *In vivo* studies on rats have demonstrated that SAC protects retinas from various insults, such as intraocular pressure augmentation and kainate excitotoxicity [143,144]. Regarding ASSNAC, *in vitro* data have shown that it can protect ARPE-19 cells from hydrogen peroxide insult, resulting in a significantly higher increase in GSH compared to cultures treated with NAC. Additionally, pre-incubation with ASSNAC has been shown to inhibit porcine lens hydrogen peroxide-induced opacification [145].

### 6.4. Riboflavin

Riboflavin (Figure 11) (vitamin B2) is a slightly water-soluble vitamin which acts as an essential cofactor in a wide range of crucial metabolic pathways such as energy production, mitochondrial function, and metabolism of other vitamins, such as B9 and B12.

Although riboflavin has not traditionally been considered an antioxidant molecule, this notion is changing because of an essential number of studies recently summarized by Olfat and colleagues, which suggest that even if indirectly, riboflavin does support the endogenous antioxidant defence system [146]. Indeed, riboflavin-deficient animals showed decreased levels of GSH, SOD, CAT, GR, and higher levels of MDA. Conversely, riboflavin administration significantly reduced MDA levels, with increased GSH production and SOD, CAT, GPXs and GR enzymes expression in different eye compartments. Moreover, riboflavin reinforces the antioxidant effects of vitamins C and E, recovering the antioxidant power of the latter through its role in the GSH redox cycle [146].

In humans, a milestone randomized trial of vitamins and minerals supplementation carried out in the rural communities of Linxian (China) (“Linxian Studies”) observed a lower prevalence of cataracts (follow-up: 5-6 years) in people receiving riboflavin together with niacin (vitamin B3) supplementation [126]. However, one limitation of this study was the lack of lens examinations at the beginning of the trial.

Due to its critical role in energy and glucose metabolism, riboflavin might play a role in DR prevention. Preliminary studies in animal models have shown that riboflavin supplementation protects the retina from oxidative stress and hyperglycaemia and increases glucose uptake and BDNF expression [147]. Moreover, riboflavin well absorbs UV light and is employed in sunlight-protective eye drops [148]. A recent study in an *in vivo* rabbit model showed a protective effect against UV-B-induced damage of an antioxidant eye drop formulation containing riboflavin and D-α-tocopheryl polyethylene glycol [149].

## 7. Defense of Cellular Membranes with Lipophilic Antioxidants

### 7.1. Vitamin E

Vitamin E (Figure 12) is a group of eight essential lipophilic substances, including tocopherols and tocotrienols, whose most active form is α-tocopherol, that protects cell membranes by breaking the chain reaction of lipid peroxidation by quenching peroxyl radicals [150]. Vitamin E function depends on the activity of other cytosolic antioxidants, such as vitamin C and GSH, as these molecules regenerate the active (reduced) form of vitamin E [120,121,151].

Vitamin E is particularly abundant in the eye in the RPE and photoreceptor cells [23]. Indeed, its deficiency leads to irreversible changes in the retinal structure and function [152].

Vitamin E (400 IU, DL-alpha tocopheryl acetate) belonged to the antioxidant mix used in the AREDS study [44]. Despite failing to counteract lens opacity of cataracts, the antioxidant mix statistically reduced the odds of developing AMD among the trial participants and was effective in preserving visual acuity, too [44,153]. On the other hand, data from other clinical reports indicate that supplementation of vitamin E alone (ranging from 100–600 IU, follow-up duration ranging from 4 to 10 years) does not prevent the development of AMD [154].

Studies investigating vitamin E supplementation for the prevention of cataracts have obtained conflicting results, too [124,154,155], suggesting a role in the pathology of the deficiency of vitamin E rather than a benefit from additional supplementation [156].

A recent study included vitamin E among those micronutrients whose deficiency is associated with DR [104].

### 7.2. Omega-3 Fatty Acids

Similarly to vitamin E, the omega-3 (ω-3) fatty acids eicosatetraenoic acid (EPA) and docosahexaenoic acid (DHA) (Figure 13) are essential nutrients for the human body. They must be introduced by nutrition (they are particularly enriched in some fish, such as salmon, mackerel, and anchovies). Once absorbed in the human body, EPA is converted to DHA via an enzymatic series of desaturation reactions.

Many studies regarding their benefits on human health have focused on the role of DHA. Omega-3 fatty acids are well known for their ability to regulate second messengers’ systems resulting in anti-inflammatory activity. In particular, *in vitro* studies suggested that it increases cellular levels of enzymatic antioxidants, such as SOD and GPXs, and augments GSH levels [157,158]. Data from human studies have also shown that DHA is a crucial element for developing the nervous system and optimal visual acuity. EPA and DHA also exhibit antioxidant properties by counteracting the accumulation of ROS and lipid peroxidation at the cellular membranes [159].

Omega-3 fatty acids are enriched in the ocular tissues, particularly the retina. Therefore, RPE physiology is strictly dependent on EPA and DHA metabolism. The latter have antioxidant effects and protective action toward retinal cells, as shown in a recent *in vitro* study performed in the ARPE-19 (a human *in vitro* model of RPE) cell line [24].

Consistently with their importance in the retina, several studies documented the efficacy of dietary supplementation of ω-3 fatty acids in various retinopathies. For example, the PREDIMED (Prevención con Dieta Mediterránea) clinical trial observed that intake of at least 500 mg/day of dietary long-chain ω-3 PUFAs (alpha-linolenic acid— ALA —, EPA, docosapentanoic acid— DPA —, and DHA) was associated with a decreased risk of DR [160].

On the other hand, data on the effects of ω-3 fatty acids on glaucoma are conflicting, and the results are linked to the dosage [161,162]. Therefore, further studies are needed to better shed light on the incidence of ω-3 fatty acids supplementation on glaucoma pathology prevention and/or progression.

The efficacy of ω-3 fatty acids in the DED has recently been reviewed by O’Byrne & O’Keeffe, revealing, despite considerable statistical heterogeneity, an improvement of subjective symptoms in DED patients [163]. On the other hand, a clinical trial (ClinicalTrials.gov Identifier: NCT01880463) was published in the same year (not included in O’Byrne & O’Keeffe meta-analysis) showing no reduction of incidence of DED by long term (median: 5.3 years) supplementation with marine ω-3 fatty acids, 1 g per day [164].

In a recent study, patients with keratoconus received DHA (1000 mg) supplementation for three months, resulting in higher total antioxidant capacity and better outcomes in the astigmatism axis, asphericity coefficient, and intraocular pressure than the placebo group [165].

The use of ω-3 fatty acids in AMD has been investigated in the AREDS2 study, concluding that the addition of DHA (350 mg) and EPA (650 mg) to the original AREDS formulation failed further to reduce the risk of progression to advanced AMD neither had an overall effect on the need for cataract surgery [45]. However, it should be noted that EPA and DHA in the AREDS2 formulation were given to subjects in the ethyl ester form. Esterification is a technological process used to increase the stability and concentration of ω-3 fatty acids from fish oils. Nonetheless, a vast amount of literature developed mainly in the last decade indicates that ω-3 fatty acids in their ethyl ester form have significantly lower bioavailability than triglyceride or monoglyceride forms [166,167,168].

Future evaluations and studies also considering the form of ω-3 fatty acids used in trials are needed better to elucidate the role of these essential nutrients in AMD.

### 7.3. Coenzyme Q10

Coenzyme Q10 (CoQ10, ubiquinone) (Figure 14) is a crucial element of cells’ electron transport chain and mitochondrial bioenergetics. Its main cellular localization is the mitochondrion, which can also be found in other subcellular fractions and plasma. Unlike other fat-soluble antioxidants, such as vitamin E and ω-3 fatty acids, CoQ10 is synthesized endogenously. Therefore, its content is superior in tissues with higher metabolic activity and oxygen consumption, such as the heart, skeletal muscle, brain, and retina. It directly regulates the redox equilibrium in the mitochondria, acting as a free radical scavenger on site. Its reduced form, ubiquinol (CoQ10H_2_), inhibits lipid peroxidation and DNA and protein oxidation [169].

Given the crucial role of CoQ10, it is not surprising that its deficiency induces apoptosis and is associated with diseases affecting the retina, such as AMD and glaucoma [170]. In this regard, CoQ10, either topically applied with eye drops or supplemented in the diet alone or in combination with vitamin E, showed evidence of decreasing retinal ganglion cell loss and slowing glaucoma progression [156]. The proposed mechanisms of action involve diminished DNA fragmentation and retinal cell apoptosis, along with a neuroprotective effect exerted by restraining extracellular glutamate accumulation, preventing cytotoxicity and pro-apoptotic factors release, indicating that CoQ10 is beyond a simple free radical scavenger [171].

Data involving CoQ10 potential role in AMD are scarce. However, one study found that CoQ10, in combination with ω-3 fatty acids and acetyl-L-carnitine, significantly improved visual field mean defect (VFMD), visual acuity and foveal sensitivity in early AMD patients [172].

Interestingly, CoQ10 is also finding application in astrobiology as a possible protector of astronauts’ eyes, which often develop neuro-ocular alteration due to the damaging effects of microgravity and cosmic radiation. For example, in an *in vitro/in-space* study, CoQ10 treatment of ARPE-19 cells transferred for three days on board the International Space Station increased cell resistance to damage from microgravity [173].

### 7.4. Alpha Lipoic Acid

α-lipoic acid (LA) (Figure 15) is a natural organosulfur with antioxidant and anti-inflammatory properties, present in animal and plant food and that can be modestly produced by the human body. It acts as a direct antioxidant, inactivating free radicals, and an indirect antioxidant, able to regenerate other cellular antiradical molecules such as GSH, vitamin C and vitamin E [174].

It has been documented that LA can cross the blood-brain barrier and, after ingestion, rapidly reaches the retina and the optic nerve [22]. This and its antioxidant properties make it a promising candidate for treating glaucoma. In a study using the DBA/2J mouse model of glaucoma, supplementation of LA in the diet of animals has been effective in both the prevention and treatment of glaucoma, assessed by reduction of RGC death as well as axon integrity and transport [175]. Despite these promising results, data on humans are scarce and further research is needed on the relationship between LA and glaucoma.

The involvement of LA as a cofactor in many energy-generating metabolic processes suggested exploiting this molecule in the treatment of complications of diabetes, including DR. In a first study, supplementation of LA in diabetic mice limited the reduction of retinal mitochondria number, a DR marker [176]. In contrast, in a second study, treatment with oral LA reduced oxidative stress, NFκB activation and VEGF in the retina of diabetic mice [177]. However, due to conflicting results in the preliminary studies, further research is required to shed light on the effect of LA on DR prevention.

LA activities in the eye have been investigated through topical application, too, despite its limited ocular penetration [178]. In a rabbit model of trabeculectomy (a surgical technique used to treat glaucoma), the instillation of 1% LA eye drops prevents fibrosis of the surgical scar by inhibiting inflammation pathways and accumulation of extracellular matrix [179]. To improve LA corneal penetration, a newly-developed micelle-based formulation Soluplus^®^ (caprolactam-polyvinyl acetate-polyethene glycol copolymer) was able to strongly increase the penetration of LA through the cornea of a bovine model [180].

## 8. Conclusions and Future Perspectives

In this review, our aim was to gather and present the key scientific findings regarding the antioxidant potential of various anatomical compartments of the eye.

The eye’s physiological processes depend on very high oxygen consumption, which, in turn, implies abundant ROS generation. Moreover, many anatomical compartments are exposed to sunlight, including highly energetic and potentially harmful wavelengths (UV light). The complex balance between radicals’ generation and scavenging is crucial in eye physiology. Losing this delicate equilibrium sets the ground for ageing and a wide range of oxidative stress-related pathologies.

Therefore, several enzymatic and non-enzymatic antioxidant molecules are deputed to safeguard and regulate the antioxidant homeostasis in the eye. From the chemical point of view, this is a heterogeneous group, including metal ions, hydrophilic and lipophilic vitamins, lipids, organic metabolites, peptides, polyphenols, and proteins (enzymes).

These substances are produced autonomously by the human body and/or are introduced from the external environment, mainly through the diet. In the latter case, these molecules are nutritionally defined as essential. Although their chemical structures can be highly diverse, non-enzymatic antioxidants have in common the ability, directly and/or indirectly, alone and/or in synergy with other antioxidants, to inactivate the radicals produced in cells by taking over the highly reactive split electrons with the ability to terminate the radical propagation chain.

From an industrial perspective, non-enzymatic antioxidants are generally easier to produce synthetically or by exploiting natural processes. Additionally, their formulation and stability in food supplements are more easily manageable compared to enzymatic antioxidants… Furthermore, given that it has been established that many eye disorders, such as DR, glaucoma, cataracts, AMD and others, have an aetiology that is rooted in an oxidative stress chronic state, it is not surprising that a large number of studies has investigated the possibility of preventing and/or attenuating these diseases with nutraceutical administration of antioxidants, with a substantial body of literature supporting this hypothesis, and presented in this review.

However, despite some promising results, a considerable number of intervention studies did not demonstrate the real efficacy of one or more oral antioxidants in preventing or attenuating a specific ophthalmic disease. Noteworthily, in many of these studies there was no assessment of the nutritional status of the participating subjects at the beginning of the experiment. The levels of a certain antioxidant at the beginning of supplementation are as essential and decisive for the output of the study as the nutritional intervention itself (dosage, follow-up, time of intervention). Therefore, we believe that studies lacking in evaluating cohorts’ “time zero” nutritional status missed an important parameter crucially contributing to the experimental output. In other words, if, in case of nutritional deficiency, the administration of the missing nutrient could prevent the onset of a disease, it is not necessarily true that the surplus of this nutrient exerts extra benefits.

Moreover, it is also essential to bear in mind that the effectiveness of antioxidants in mitigating oxidative stress also depends on the complex synergistic network with other antioxidant molecules, which can cooperate in complex redox cycles and/or regenerate the activity of each other. This is the case, for example, of glutathione, one of the most important endogenous antioxidants in the human body, which requires the co-presence of other antioxidants, such as vitamins B2, C and E, to carry out its activity and glutathione is synthesized using N-acetyl-cysteine as one of its substrates. Therefore, researchers in the nutraceutical supplementation field will face the challenge of designing future clinical studies that incorporate the aforementioned considerations. This includes evaluating the effects of specific nutrients on health outcomes from a more objective and synergistic perspective.

Another important observation that must be highlighted is that the antioxidant molecules presented in this review have antioxidant actions and diverse and varied biological roles. Some non-exhaustive examples are omega-3 fatty acids, which have an essential anti-inflammatory role and intervene in retinal function via second messenger regulation, CoQ10 and lipoic acid, which are essential cofactors of energy metabolism, as well as the many and diverse cofactor roles of micronutrients (vitamins and minerals) in a vast array of biological pathways. Thus, in considering the efficacy of a molecule with antioxidant properties in modulating the odds of onset and/or progression of a disease, its antioxidant activity plays an important role. However, it is not the only way to explain the molecule’s efficacy.

Finally, an aspect of fundamental importance in the administration (oral or topical) of the molecules mentioned here is their bioavailability, i.e., the amount of the administered molecule that reaches the site of action. We are aware that the bioavailability of many of these molecules can be an issue in their delivery. Indeed, the complex structure of the eye poses a series of anatomical barriers, such as the blood-brain barrier and the fact that the eye’s anterior chamber is partly not vascularised. However, given the importance of this topic, we decided not to address it here in-depth, as it deserves to be approached separately.

In light of the scientific literature produced so far, we believe that the role of antioxidant molecules in eye health is still underestimated and that future studies may highlight their potential and give rise to new applications in terms of nutritional strategies as well as medical devices and pharmaceutical formulations.

## Figures and Tables

**Figure 1 nutrients-15-02283-f001:**
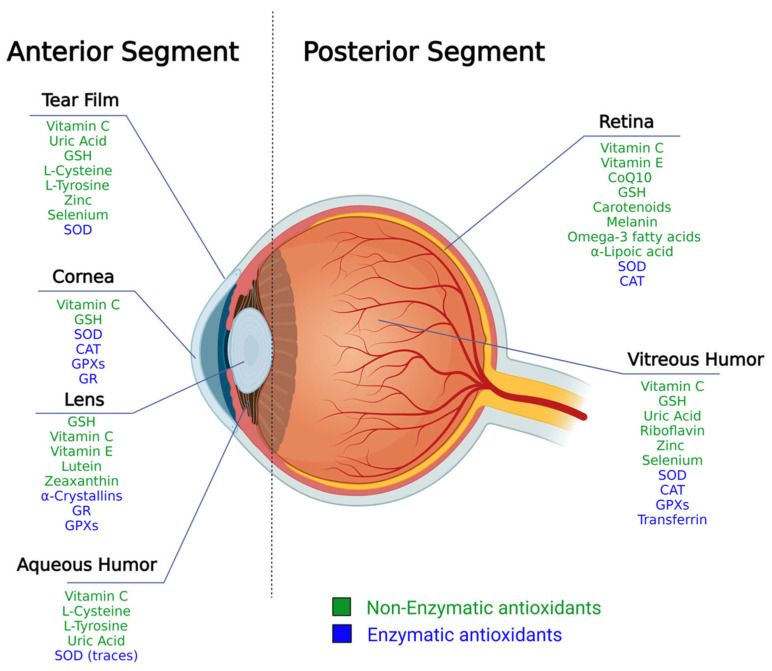
Enzymatic (blue) and non-enzymatic (green) antioxidant molecules are identified in the different compartments of the eye. CAT: catalase; CoQ10: coenzyme Q10; GPXs: glutathione peroxidases; GR: Glutathione Reductase; GSH: Glutathione; SOD: superoxide dismutase. Reference for figure content: [7,8,9,10,11,12,13,14,15,16,17,18,19,20,21,22,23,24,25] created with BioRender.com.

**Figure 2 nutrients-15-02283-f002:**
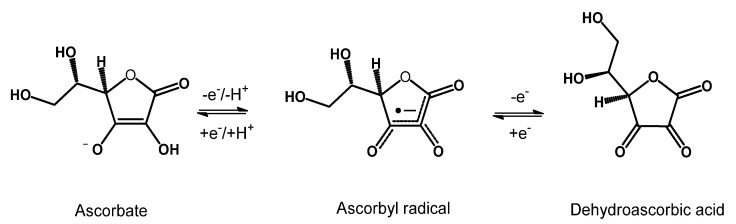
Oxidation of ascorbate by two successive one-electron oxidation steps to give the ascorbyl radical and dehydroascorbic acid.

**Figure 3 nutrients-15-02283-f003:**
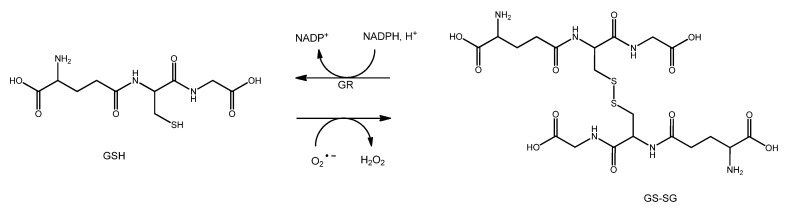
Glutathione oxidation/reduction.

**Figure 4 nutrients-15-02283-f004:**
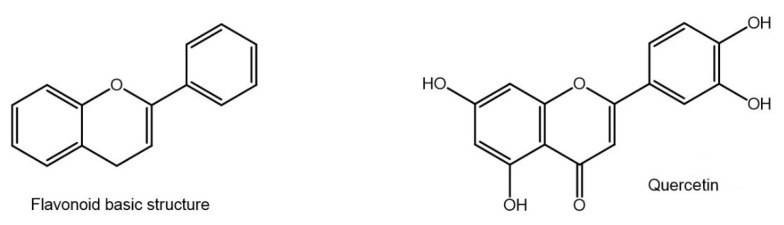
Basic flavonoid backbone and quercetin structure.

**Figure 5 nutrients-15-02283-f005:**
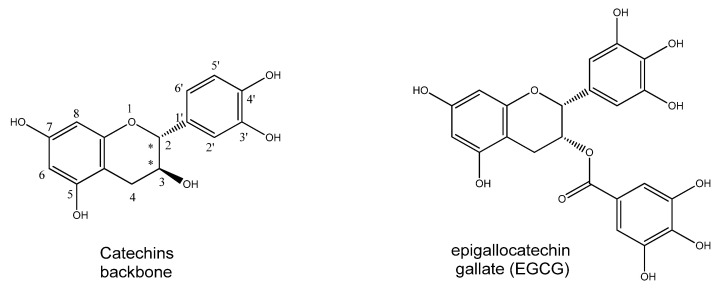
Catechin backbone, where stars underline chiral carbons, and epigallocatechin gallate structure.

**Figure 6 nutrients-15-02283-f006:**
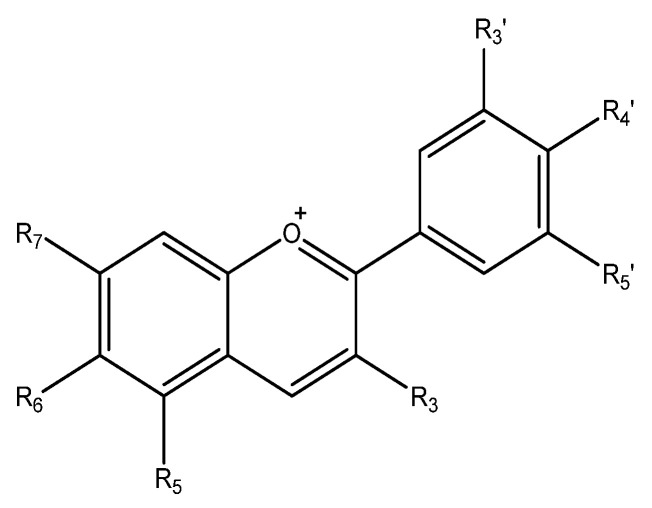
Basic anthocyanin structure.

**Figure 7 nutrients-15-02283-f007:**
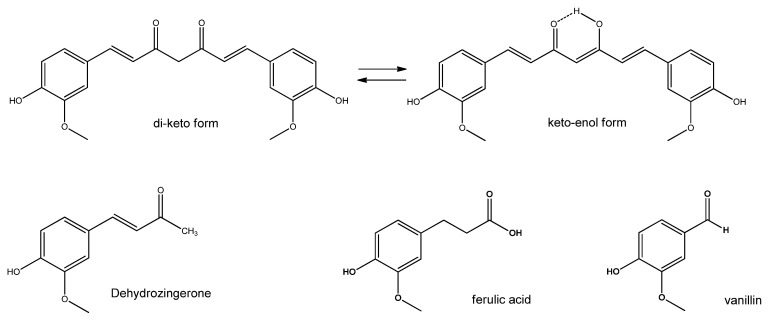
Keto-enol tautomerism of curcumin and metabolism byproducts.

**Figure 8 nutrients-15-02283-f008:**
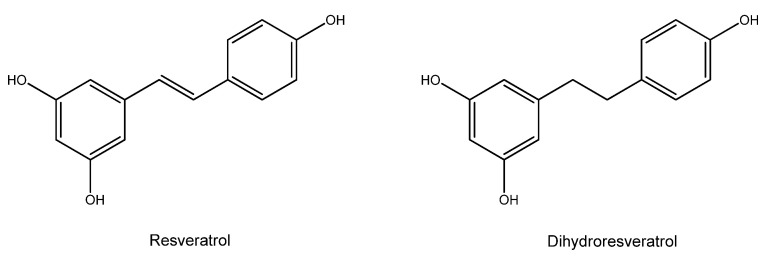
Resveratrol and its metabolite dihydroresveratrol.

**Figure 9 nutrients-15-02283-f009:**
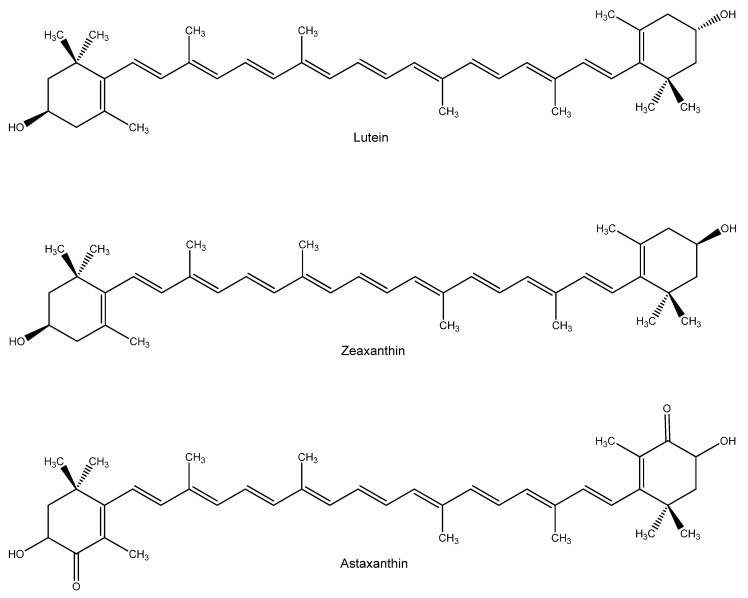
Structure of lutein, zeaxanthin and astaxanthin.

**Figure 10 nutrients-15-02283-f010:**
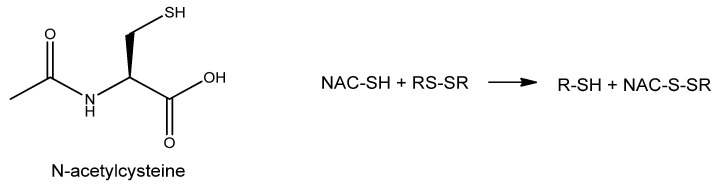
N-acetyl cysteine structure and its disulphide bonds reduction reaction.

**Figure 11 nutrients-15-02283-f011:**
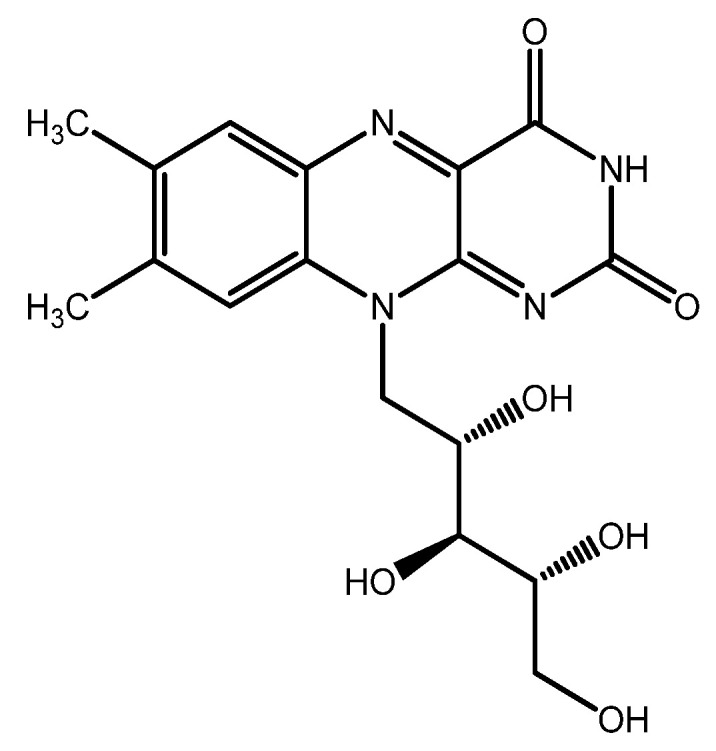
Riboflavin structure.

**Figure 12 nutrients-15-02283-f012:**
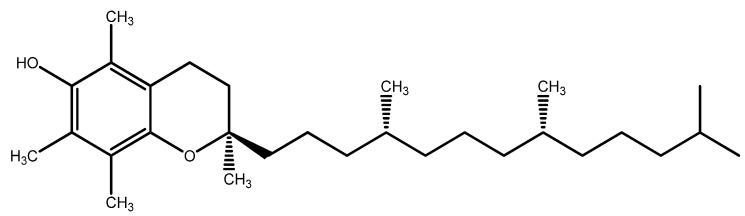
Vitamin E structure.

**Figure 13 nutrients-15-02283-f013:**
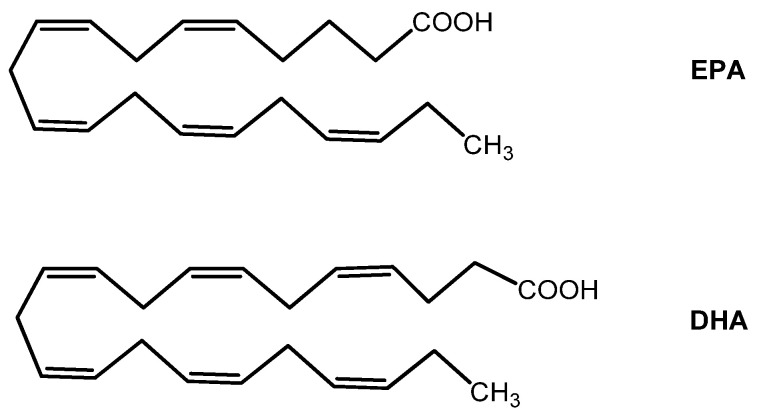
Eicosapentaenoic acid (EPA) and docoahexanoic acid (DHA) structures.

**Figure 14 nutrients-15-02283-f014:**
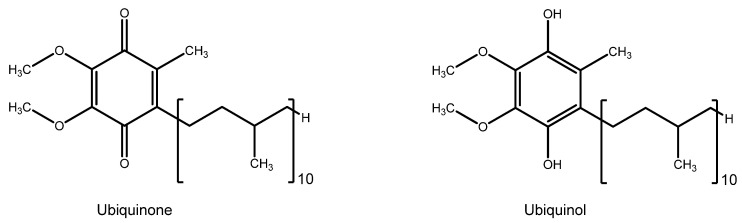
Ubiquinone and ubiquinol structure.

**Figure 15 nutrients-15-02283-f015:**
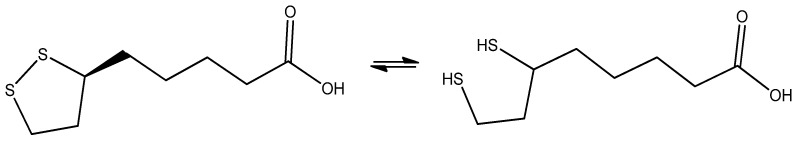
Alpha-lipoic acid is a cyclic disulfur in equilibrium with the linear reduced form dihydrolipoic acid.

**Table 1 nutrients-15-02283-t001:** The primary antioxidants are used in food supplements and ophthalmic medical devices. EPA: eicosapentaenoic acid; DHA: docosahexaenoic acid; GSH: glutathione; ROS: reactive oxygen species; AREDS [44,45,46] and AREDS2 [47,48]: Age-Related Eye Disease Study (1 and 2).

Group	Antioxidant	Mechanism(s) of Antioxidant Action	Notes
**Polyphenols**	**Flavonoids** **Catechins** **Anthocyanins**	- Radical scavengers - Metal chelators - Inhibition of ROS-generation enzymes - Expression of antioxidant enzymes - Anti-inflammatory	
	**Curcumin**	- Radical scavenger - Metal Chelator - Anti-inflammatory - Anti-angiogenetic	
	**Resveratrol**	- Radical scavenger - Pro-oxidant effects, beneficial in cancer - Expression of antioxidant enzymes - Anti-inflammatory	
**Carotenoids**	**Beta-carotene**	- Cell membrane antioxidant - Regeneration of other antioxidants - Vitamin A precursor	- Included in the AREDS study
	**Lutein**	- Cell membrane antioxidant - Regeneration of other antioxidants	- Xanthophylls subgroup - included in the AREDS2 study
	**Zeaxanthin**	- Cell membrane antioxidant - Regeneration of other antioxidants	- Xanthophylls subgroup - included in the AREDS2 study
	**Astaxanthin**	- Cell membrane antioxidant - Direct antioxidant - Anti-inflammatory - Neuroprotective	- Xanthophylls subgroup
**Water-Soluble Promoters of Endogenous Antioxidant System**	**Zinc**	- A building block for redox system enzymes	- Included in the AREDS study
	**Selenium**	- A building block for redox system enzymes	
	**Vitamin C**	- Direct antioxidant - Synergy with other antioxidants (Vitamin E, GSH, flavonoids)	- Included in the AREDS study - The most abundant antioxidant in the eye
	**N-acetyl** **cysteine**	- Protection from sulphydryl oxidation - Scavenger of superoxide and peroxide radicals - Precursor of GSH	
	**Riboflavin**	- Indirect antioxidant (supports the endogenous antioxidant system) - UV light absorption	
**Lipophilic Antioxidants in cellular membranes**	**Vitamin E**	- Cell membrane antioxidant - Synergy with other antioxidants (Vitamin C, GSH) - UV light absorption	- Included in the AREDS study
	**Omega-3 fatty acids EPA and DHA**	- Cell membrane antioxidant - Anti-inflammatory	- Included in the AREDS2 study - Nervous system and retina development
	**Coenzyme Q10**	- Mitochondrial redox equilibrium regulator and antioxidant - Neuroprotective	- Energy metabolism
	**Alpha Lipoic Acid**	- Direct antioxidant - Regeneration of other antioxidants - Anti-inflammatory	- Energy metabolism

## Data Availability

Not applicable.

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
