# Peer review of "Antioxidant Nutraceutical Strategies in the Prevention of Oxidative Stress Related Eye Diseases"

_nutrients, 2023, doi:10.3390/nu15102283_

Round 1

Reviewer 1 Report

I believe that valuable information has been collected and summarised in s.

Many researchers are interested in antioxidants and incorporate them into their research agenda.

1)Well, 6.2. Ascorbic Acid mentions water-soluble ascorbic acid.

If possible, could you add other water-soluble ascorbic acid derivatives (e.g. AA2G)?

Can the liposoluble ascorbic acid derivatives (e.g. ascorbic acid palmitate) also be referred to?

2)With regard to '6.3. N-Acetyl cysteine', could you add the relationship with SAC (S-allyl cysteine), e.g. metabolic processes?

3)Finally, why not add a message to researchers who are interested in antioxidant research as a challenge?

I am not concerned about the English text. Again, please check for TYPOn.

Author Response

REVIEWER 1

Comments and Suggestions for Authors

I believe that valuable information has been collected and summarised in s.

Many researchers are interested in antioxidants and incorporate them into their research agenda.

 We thank the reviewer for the general appreciation.

1)Well, 6.2. Ascorbic Acid mentions water-soluble ascorbic acid.

If possible, could you add other water-soluble ascorbic acid derivatives (e.g. AA2G)?

We thank the reviewer for suggestion. Ophthalmological applications of AA-2G are presented in chapter 6.2 (lines 543-553).

Can the liposoluble ascorbic acid derivatives (e.g. ascorbic acid palmitate) also be referred to?

Unfortunately, we have not been able to adequately include information on ascorbyl palmitate, as there are (to our knowledge) no studies in the literature that have evaluated the effects of ascorbyl palmitate in the ophthalmic field.

2)With regard to '6.3. N-Acetyl cysteine', could you add the relationship with SAC (S-allyl cysteine), e.g. metabolic processes?

Thanks for suggestion. We have changed the title of chapter 6.3 in “N-Acetyl cysteine' and other cysteine derivatives” adding the ophthalmological applications of SAC and also of S-Allylmercapto-N-acetylcysteine (ASSNAC). (lines 592-600)

  ï¼“)Finally, why not add a message to researchers who are interested in antioxidant research as a challenge?

We agree with the comment.

Indeed we believe that, for the future, clinical studies investigating the effects of nutraceutical supplmentation on health outputs shall adress two crucial issues:

  1. The “time zero” nutritional status of a given nutrient in the population before the start of the supplementation (lines 801-809).
  2. The fact that the effect of a nutraceutical depends also in the nutritional status of other nutrients, in a synergistic network view of the efficacy of nutraceutical supplementation (lines 810-816).

Accordingly, an important challenge for researchers in the antioxidant field will be to project their experimental designs adressing what abovementioned.

A comment to the manuscript (lines: 817-820) has been added in order to underline these considerations.

Comments on the Quality of English Language

I am not concerned about the English text. Again, please check for TYPOn.

Submission Date

14 April 2023

Date of this review

20 Apr 2023 09:02:42

Reviewer 2 Report

The topic of the article is very relevant since alternatives are currently being sought to help improve the health conditions of the population. In this sense, it is very timely to have information associated with nutraceutical compounds that can help with problems associated with oxidative stress and improve the visual problems that occur.

The structure of the article and the information presented is clear and well-documented, with a significant number of supporting articles. I want to highlight as authors leading the reader to understand the structure of the eye and ocular antioxidant defense system, joining with diseases and their possible causes, making it an article for any reader.  It also relates each of the nutraceutical compounds to their beneficial effects on eye health, making it very interesting. 

Author Response

REVIEWER 2

Comments and Suggestions for Authors

The topic of the article is very relevant since alternatives are currently being sought to help improve the health conditions of the population. In this sense, it is very timely to have information associated with nutraceutical compounds that can help with problems associated with oxidative stress and improve the visual problems that occur.

The structure of the article and the information presented is clear and well-documented, with a significant number of supporting articles. I want to highlight as authors leading the reader to understand the structure of the eye and ocular antioxidant defense system, joining with diseases and their possible causes, making it an article for any reader.  It also relates each of the nutraceutical compounds to their beneficial effects on eye health, making it very interesting. 

We really thank the reviewer for the appreciation. Our aim was indeed to provide useful information even to those not in the ophthalmological field.

Submission Date

14 April 2023

Date of this review

29 Apr 2023 01:47:11